# Clinical Outcome of Nivolumab Plus Ipilimumab in Patients with Locally Advanced Non-Small-Cell Lung Cancer with Relapse after Concurrent Chemoradiotherapy followed by Durvalumab

**DOI:** 10.3390/cancers16071409

**Published:** 2024-04-03

**Authors:** Atsuto Mouri, Satoshi Watanabe, Takaaki Tokito, Yoshiaki Nagai, Yu Saida, Hisao Imai, Ou Yamaguchi, Kunihiko Kobayashi, Kyoichi Kaira, Hiroshi Kagamu

**Affiliations:** 1Department of Respiratory Medicine, International Medical Center, Saitama Medical University, Hidaka 350-1298, Japan; mouria@saitama-med.ac.jp (A.M.); m06701014@gunma-u.ac.jp (H.I.); ouyamagu@saitama-med.ac.jp (O.Y.); kobakuni@saitama-med.ac.jp (K.K.); kkaira@saitama-med.ac.jp (K.K.); kagamu19@saitama-med.ac.jp (H.K.); 2Department of Respiratory Medicine and Infectious Diseases, Niigata University Graduate School of Medical and Dental Sciences, Niigata 951-8510, Japan; saida@med.niigata-u.ac.jp; 3Division of Respirology, Neurology, and Rheumatology, Department of Internal Medicine, Kurume University School of Medicine, Kurume 830-0011, Japan; tokitou_takaaki@kurume-u.ac.jp; 4Division of Respiratory Medicine, Clinical Department of Internal Medicine, Jichi Medical University Saitama Medical Center, Saitama 330-0834, Japan; nagai@saitama-med.ac.jp

**Keywords:** nivolumab, ipilimumab, durvalumab, CCRT, non-small-cell lung cancer

## Abstract

**Simple Summary:**

The clinical significance of nivolumab plus ipilimumab in patients with recurrence after chemoradiotherapy followed by durvalumab remains unclear. We investigated the efficacy and safety of nivolumab plus ipilimumab in patients who relapsed after chemoradiotherapy followed by durvalumab. The 6- and 12-month PFS rates for nivolumab plus ipilimumab for relapse after durvalumab were 46.7% and 36.4%, indicating a long-tail plateau. Nivolumab plus ipilimumab was particularly effective in patients treated with durvalumab for 6 months or more. Therefore, nivolumab plus ipilimumab is a promising treatment option for patients who have relapsed after CCRT–durvalumab, with good tolerability.

**Abstract:**

Nivolumab plus ipilimumab showed promising efficacy in patients with metastatic non-small-cell lung cancer (NSCLC). The efficacy of the nivolumab plus ipilimumab combination regimen in NSCLC patients who relapse after durvalumab consolidation following concurrent chemoradiotherapy (CCRT) has not been determined. Between January 2021 and June 2022, clinical data were retrospectively extracted from the medical records of patients with NSCLC who received nivolumab plus ipilimumab after CCRT and durvalumab consolidation. A total of 30 patients were included in this analysis. The median number of durvalumab treatment cycles was 11. Median PFS and OS with nivolumab plus ipilimumab were 4.2 months (95% confidence interval [CI]: 0.7–7.7) and 18.5 months (95% CI: 3.5–33.5), respectively. The 6-month and 12-month PFS rates were 46.7% (95% CI: 28.8–64.5) and 36.4% (95% CI: 19.0–53.7). In multivariate analysis, a significant correlation was observed between a durvalumab treatment duration of 6 months or more and PFS (*p* = 0.04) as well as OS (*p* = 0.001). Grade 3 adverse events, including pneumonitis, dermatitis, and colitis, occurred in 10% of the patients. This study suggests that nivolumab plus ipilimumab is effective, especially in patients who have received durvalumab for 6 months or more, and tolerable for patients who relapsed after durvalumab following CCRT.

## 1. Introduction

Immune checkpoint inhibitors (ICIs) are utilized as the standard systemic therapy in various types of cancer [1,2]. In patients with non-small-cell lung cancer (NSCLC), ICIs help achieve clinical efficacy as a first-line treatment, perioperative adjuvant or neoadjuvant therapy, and as a maintenance agent after concurrent chemoradiotherapy (CCRT) [3,4,5,6,7,8]. Durvalumab, a programmed cell death-ligand 1 (PD-L1) inhibitor, as consolidation therapy after CCRT, is standard care for patients with unresectable locally advanced NSCLC (LA-NSCLC), exhibiting a survival benefit of 5-year overall survival (OS) rate of 42.9% [9]. However, approximately two-thirds of patients who receive durvalumab are not curable, requiring a different treatment after relapse.

Anti-PD-(L)1 and anti-cytotoxic T-lymphocyte-associated antigen 4 (CTLA-4) antibodies enhance anti-tumor immune responses through different phases and mechanisms [10]. PD-1 binds to the ligands PD-L1 and PD-L2 and inhibits T-cell proliferation, and production of interferon-γ, tumor necrosis factor-α, and IL-2, and reduces T-cell survival [11]. Some T cells with PD-1 expression called “exhausted” T cells experience high levels of stimulation or reduced CD4^+^ T-cell help [12]. CTLA-4 is a classical immune checkpoint molecule that acts as a CD28 homolog and binds with high affinity to its receptor B7-1 (CD80) or B7-2 (CD86). Unlike CD28, binding of CTLA-4 to B7 is not able to produce a stimulatory signal [13]. This competitive binding prevents the costimulatory signal provided by CD28 and B7-1/2 binding. Moreover, several immune checkpoint targets including T-cell immunoglobulin and ITIM domain, and lymphocyte activation gene-3, like T-cell immunoglobulin and mucin-domain containing-3, have been discovered and are being investigated in clinical research. However, only ICIs targeting PD-1/PD-L1 and CTLA-4 have been approved and widely used in various cancers. The combination therapy of nivolumab and ipilimumab demonstrated favorable 5-year survival rates even in patient populations with PD-L1 tumor proportion scores of less than 1%, where PD-1/PD-L1 inhibitors are considered less effective [14]. Moreover, previous studies have demonstrated the clinical benefits of PD-1 inhibitor plus ipilimumab in patients with metastatic melanoma who experienced resistance to PD-(L)1 inhibitor [15]. Among patients with recurrence after durvalumab consolidation therapy following CCRT (CCRT–durvalumab), some patients did not achieve sufficient enhancement of anti-tumor immunity with anti-PD-L1 monotherapy. For these patients, the combination of nivolumab plus ipilimumab might be potentially effective. However, the clinical significance of nivolumab plus ipilimumab in patients with recurrence after CCRT–durvalumab remains unclear.

Based on these backgrounds, this retrospective study aimed to determine the clinical efficacy of nivolumab plus ipilimumab in patients with LA-NSCLC who relapsed after CCRT–durvalumab.

## 2. Materials and Methods

### 2.1. Patients

This multicenter, retrospective cohort study enrolled patients with LA-NSCLC who were treated with a nivolumab plus ipilimumab combination regimen due to cancer recurrence after CCRT–durvalumab treatment. The patients were evaluated at any of the four participating institutions in Japan and received nivolumab and ipilimumab regardless of whether chemotherapy was or was not concomitant, between January 2021 and June 2022. The eligibility criteria were as follows: (i) pathologically diagnosed NSCLC and (ii) progression after treatment with definitive CCRT (curative intent followed by consolidation durvalumab). Recurrence was defined as an apparent worsening on computed tomography scanning with or without histopathological proof of existing lung cancer. The local recurrence was defined as either the progression of an ipsilateral lung lesion contiguous with the primary tumor or a single mediastinal/hilar/supraclavicular lymph node metastasis. The clinical stages of all patients were classified according to the eight editions of the TNM classification system [16]. PD-L1 expression level was determined by the TPS, which was reported as a percentage on a scale of 0% to 100%. The TPS was evaluated using the Pharm Dx 22C3 PD-L1 assay (Agilent, Santa Clara, CA, USA).

This study was approved by the Clinical Research Review Board of Saitama Medical University (2022-018) and conducted in accordance with the principles of the Declaration of Helsinki. Informed consent was not required because of the retrospective nature of the study.

### 2.2. Treatment and Follow-Up Evaluation

Durvalumab was administrated every 2 weeks at a dose of 10 mg/kg as consolidation therapy after CCRT until disease progression or unacceptable toxicity or discontinuation as determined by the treating physician for up to 12 months. Nivolumab at a dose of 240 mg/day every 2 weeks or 360 mg/day every 3 weeks was administered intravenously. Ipilimumab at a dose of 1 mg/kg was administered intravenously every 6 weeks until disease progression, unacceptable toxicity, or discontinuation as determined by the treating physician. When chemotherapy was used in combination with nivolumab and ipilimumab, the regimen consisted of carboplatin with an area under the curve (AUC) of 5, along with pemetrexed at 500 mg/m^2^ for non-squamous cell carcinoma, or carboplatin with an AUC of 5 along with paclitaxel at 200 mg/m^2^ for squamous cell carcinoma.

Any toxicity was assessed and graded using the Common Terminology Criteria for Adverse Events version 5.0. Treatment efficacy was evaluated by the attending physician and radiologist using the Response Evaluation Criteria in Solid Tumors version 1.1 [17]. Anonymized clinical data were recorded by local investigators using case report forms and stored in a password-protected secure system. The primary endpoint was to assess the progression-free survival (PFS) and PFS rate at 6 months as measured from the first administration of nivolumab and ipilimumab to progression or death from any cause. The secondary endpoints included OS, objective response rate (ORR), and safety. The analysis of PFS, OS, and ORR was conducted using the data cut-offs that were employed for the primary analysis of PFS.

### 2.3. Statistical Analyses

PFS and OS were estimated using the Kaplan–Meier method. Univariate and multivariate analyses according to baseline characteristic subgroups were performed using Cox regression analysis. The statistical significance level was set at *p*-values of <0.05. All statistical analyses were performed using Microsoft Excel 2019 (Microsoft Corporation, Redmond, WA, USA) and JMP 14.0 (SAS Institute Inc., Cary, NC, USA).

## 3. Results

### 3.1. Characteristics of Patients and Treatment Information

A total of 30 patients were included in further analysis. At the data cut-off (31 December 2022), the median follow-up time was 10.3 months (range, 1.4–23.3). Patient characteristics are shown in Table 1. The median age of the patients was 72 years (range, 50–80) and there were 29 (96.7%) males. Fifteen patients (50%) had lung squamous cell carcinoma, 14 (46.7%) had adenocarcinoma, and 1 (3.3%) was diagnosed as not otherwise specified. Twenty-nine patients (96.7%) had an Eastern Cooperative Oncology Group performance status of 0 or 1. The TPS based on PD-L1 immunohistochemistry was <1% in 9 (30%) patients, 1–49% in 14 (46.7%) patients, and ≥50% in 5 (16.7%) patients. Of 30 patients, driver mutations and rearrangements, including *EGFR*, *ALK*, and *ROS1*, were not identified in 25 patients. Additionally, five patients did not undergo driver oncogene testing. The median number of administrations of durvalumab was 10.5 (range, 1–26) and the median treatment duration with durvalumab was 5.8 months (range, 0.03–11.9). The median time from completion of irradiation to recurrence was 6.8 months (range, 1.7–30.7). Of 30 patients, 26 patients received nivolumab plus ipilimumab without chemotherapy (CheckMate 227 regimen), while 4 patients received nivolumab plus ipilimumab in combination with platinum-doublet chemotherapy (CheckMate 9LA regimen). Detailed information on CCRT–durvalumab including chemotherapy regimen are shown in Appendix A.

### 3.2. Efficacy and Survival

The median PFS and OS with nivolumab plus ipilimumab were 4.2 months (95% CI: 0.7–7.7) and 18.5 months (95% CI: 3.5–33.5), respectively. The median time to treatment failure was 5.4 months (95% CI: 0–11.1) (Figure 1). The 6-month PFS rate was 46.7% (95% CI: 28.8–64.5), and the 12-month PFS rate was 36.4% (95% CI: 19.0–53.7). After nivolumab plus ipilimumab treatment, partial response was observed in 7 patients, stable disease in 14 patients, and progression of disease in 9 patients. The ORR was 23.3% (95% CI: 12–41), and the disease control rate was 70% (95% CI: 52–83) (Table 2).

The results of the univariate and multivariate analyses for PFS and OS are summarized in Table 3. Three factors had a prognostic significance in univariate analysis for PFS: treatment cycles of durvalumab (*p* = 0.005), treatment duration of durvalumab for 6 months or more (*p* = 0.005), and time to recurrence from final CCRT (*p* = 0.04). Likewise, three were identified as having prognostic significance in univariate analysis for OS: treatment cycles of durvalumab (*p* = 0.005), dose completion of durvalumab (*p* = 0.02), and treatment duration of durvalumab for 6 months or more (*p* = 0.005). As the factors related to durvalumab administration were similar, a multivariate analysis was conducted by selecting relevant factors with low *p*-values to mitigate multicollinearity. Duration of durvalumab treatment for 6 months or more was an independent prognostic factor for both PFS (hazard ratio 0.3 [95% CI: 0.1–1], *p* = 0.04) and OS (hazard ratio 0.1 [95% CI: 0.02–0.4], *p* = 0.0007). Appendix A shows swimmer plots illustrating the treatment duration with durvalumab and nivolumab plus ipilimumab. Of 30 patients, 17 patients were undergoing treatment with nivolumab and ipilimumab.

For further analysis, patients were divided into three groups: those who progressed after less than 6 months of treatment with durvalumab, those who progressed after 6–12 months, and those who relapsed after 12 months with completion of durvalumab. The Kaplan–Meier curves are shown in Appendix A. Compared to a median PFS of 3.5 months (95% CI: 1.5–6.4) for the 14 patients who relapsed less than 6 months after completion of radiation, the PFS for those who relapsed at 6–12 months was not reached (95% CI: 0.3 not reached, *p* = 0.10), and for the group of patients who completed 12 months of durvalumab, the PFS was also not reached (95% CI: 2.1 not reached, *p* = 0.09). Compared to a median OS of 8.2 months (95% CI: 3.3 not reached) for the patients who relapsed less than 6 months after completion of radiation, the OS for those who relapsed at 6–12 months was 10.7 months (95% CI: 3 not reached, *p* = 0.61), and for the group of patients who completed 12 months of durvalumab, the OS was 18.5 months (95% CI: 3.6 not reached, *p* = 0.08).

### 3.3. Safety

Treatment safety profiles assessed by common terminology criteria for immune-related adverse events (irAEs) are presented in Table 4. The most common irAEs were liver dysfunction (20%), skin disorders (20%), hyperthyroidism or hypothyroidism (16.7%), pneumonitis (10%), arthralgia (10%), and diarrhea (6.7%). Grade 3 irAEs included pneumonitis (3.3%), skin disorders (3.3%), and diarrhea (3.3%). None of the patients experienced treatment-related deaths. There were no reports of unexpected irAEs.

The AEs during durvalumab administration are listed in Appendix A. To compare AEs during durvalumab and during nivolumab plus ipilimumab treatment, the list of AEs for each patient is shown in Appendix A. The same AEs occurred in 26.7% (8/30) of patients, of which 2 patients had a worse grade with nivolumab plus ipilimumab than with durvalumab (Figure 2).

## 4. Discussion

The present study revealed the moderate efficacy of nivolumab plus ipilimumab in patients with LA-NSCLC relapse after CCRT–durvalumab treatment. Our study referenced a previous report that indicated that 24-week progression is the most accurate predictor of survival in NSCLC patients treated with ICIs [18]. The CheckMate 227 trial investigating nivolumab and ipilimumab in metastatic NSCLC patients showed a 6-month PFS rate of 40–50% [14]. In our study, the primary endpoint of the 6-month PFS rate was 46.7%, indicating a survival benefit of nivolumab plus ipilimumab even after CCRT–durvalumab. Of note, nivolumab plus ipilimumab resulted in a 12-month PFS rate of 36.4%. In addition, NSCLC patients who had received durvalumab for 6 months or more had significantly better PFS and OS. To the best of our knowledge, this is the first study to show that the combination of nivolumab and ipilimumab demonstrated survival benefits in patients with relapse after CCRT–durvalumab.

Tremendous attention has been paid to the optimal treatment for patients experiencing relapse after CCRT–durvalumab. Other treatments for this population include cytotoxic agents, platinum-based regimens, or a combination of those plus PD-(L)1 inhibitor [19]. Imai et al. showed that the platinum-based regimen resulted in a response rate of 16.7% and PFS of 4.2 months for patients who relapsed after chemoradiotherapy without durvalumab [20]. Miyawaki et al. discussed the benefit of platinum-based chemotherapy in similar patients without durvalumab [21]. The current study showed that the efficacy of nivolumab and ipilimumab is comparable to or better than those studies, despite cases of relapse after durvalumab. Amino et al. reported a favorable response rate of 45% for the anti-PD-1 antibody regimen in 20 patients with recurrence after CRT without durvalumab in stage III NSCLC [22]. Nonetheless, based on the available evidence, it appears that the re-challenge of PD-(L)1 inhibitors offers limited benefits [23]. Simultaneous inhibition of both PD-(L)1 and CTLA-4 could restore the anti-tumor immune response in patients previously treated with either PD-1 or CTLA-4 inhibitors alone. In malignant melanoma, the addition of CTLA-4 inhibitor after PD-1 inhibitor may improve PFS rates [15,24]. In both malignant melanoma and lung cancer, Kaplan–Meier curves from long-term follow-up analyses showed that the nivolumab plus ipilimumab therapy was superior to nivolumab alone with or without chemotherapy [14,25,26]. Considering the factors that contributed to the favorable outcome of the combination of CTLA-4 in this study, it is possible that the anti-tumor immune responses were activated because all patients were post-irradiation [22]. In addition, the selection bias, in which the attending physician included patients who were expected to benefit from nivolumab plus ipilimumab, should also be considered.

In the present study, patients who relapsed after the completion of durvalumab treatment tended to benefit from nivolumab plus ipilimumab, suggesting that these patients may have had cancer reactivation due to the ICI treatment discontinuation. This study could not clarify whether PD-1 inhibitor plus CTLA-4 inhibitor or PD-(L)1 inhibitor alone was sufficient to re-initiate or maintain the anti-tumor effect in such patients. Meanwhile, no significant differences in survival were associated with PD-L1 expression level, metastatic disease, or recurrence after the end of durvalumab treatment. Consistent with previous reports, distant metastases were the most common form of recurrence in this study [27]. It was assumed that the local recurrence cases included a population in which the anti-tumor immune system was relatively active in suppressing extensive progression, such as distant metastases. However, the results of the current study did not indicate that patients with local recurrence had a significantly better prognosis.

Nivolumab plus ipilimumab combination regimens for patients who had experienced radiation therapy did not appear to increase toxicity compared with the CheckMate 227 trial [7,8]. Of note, eight patients in the current study had the same AEs as during durvalumab treatment, and two patients had noticeable grade worsening compared to during durvalumab treatment, including skin disorder and pneumonitis of grade 2. Although grade 3 or 4 pneumonitis, dermatitis, and colitis were identified in the current study, the short follow-up period may have led to an underestimation of AEs. Treatment-related grade 3–4 AEs were invariably higher compared to anti-PD-1 antibodies alone in any reports [28,29]. Therefore, more attention should be paid when prescribing treatment with nivolumab plus ipilimumab.

Several limitations of our study warrant mention. First, it was a retrospective study with a small size. Therefore, the reported values may be under- or overestimated. Large trials are required to examine whether parameters such as PD-L1 < 1% and the pattern of local recurrence can predict treatment benefits. Four patients treated with the CheckMate 9LA regimen were included in this study. Compared to CheckMate 227 regimen cases, they had similar PFS and OS without significant differences (data not shown). We believe that the CheckMate 9LA regimen may be preferred in patients with large tumor volume or rapid disease progression, and this regimen could be a treatment option for patients with recurrence after CCRT–durvalumab. This study suggested the potential efficacy of nivolumab plus ipilimumab combination therapy in patients with recurrent NSCLC following durvalumab consolidation therapy. However, it should be noted that adherence to clinical guidelines and regulatory agency policies is imperative when considering its clinical use. Second, the advantages of ICIs for long-term outcomes were not evaluated. Although the Pacific study demonstrated a significant improvement in survival with durvalumab consolidation therapy, approximately 45% of patients experienced recurrence after CCRT–durvalumab within one year [9]. Therefore, we believe it is advisable to generate data demonstrating the efficacy of nivolumab plus ipilimumab in these patients as soon as possible. Although the observation period was short, a tail plateau was observed with this treatment modality, and early reporting was a priority. Third, in addition to PD-L1, other tumor biomarkers, such as gene mutation burden and tumor microenvironment were not evaluated. Some biomarkers may help determine whether the combination of nivolumab and ipilimumab or anti-PD-(L)1 antibody alone is optimal for patients who relapse after the completion of durvalumab treatment [30]. In addition, sensitive predictive biomarkers are required to avoid ICI overdose, which could lead to the exposure of potential non-responders to unnecessary immunotoxicity. In the current study, PD-L1 expression on tumor cells was evaluated with 22C3 at diagnosis with NSCLC. Although in previous clinical trials investigating durvalumab, PD-L1 expression in tumor microenvironment was evaluated using an SP263 clone, the high correlation between PD-L1 expression data was obtained with 22C3 and SP263 [31]. Because of the popularity of 22C2 in PD-L1 testing, PD-L1 expression has usually been assessed with 22C3 in Japan.

## 5. Conclusions

Nivolumab plus ipilimumab was identified as a feasible and efficient treatment for patients with LA-NSCLC relapsing after CCRT–durvalumab. Nivolumab plus ipilimumab treatment may be particularly effective in patients who received durvalumab for 6 months or more. Prospective comparative trials between nivolumab plus ipilimumab therapy and cytotoxic anticancer agents with or without PD-(L)1 therapy are desirable.

## Figures and Tables

**Figure 1 cancers-16-01409-f001:**
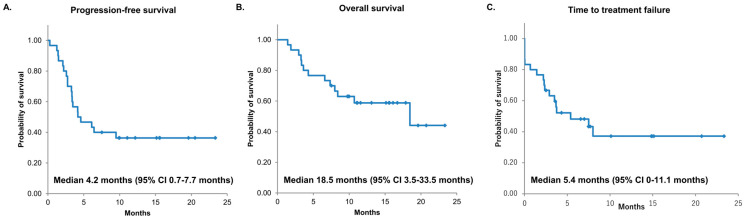
Kaplan–Meier survival curves of PFS (**A**), OS (**B**), and TTF (**C**). PFS, progression-free survival; OS, overall survival: TTF, time to treatment failure: CI, confidential interval; and NR, not reached.

**Figure 2 cancers-16-01409-f002:**
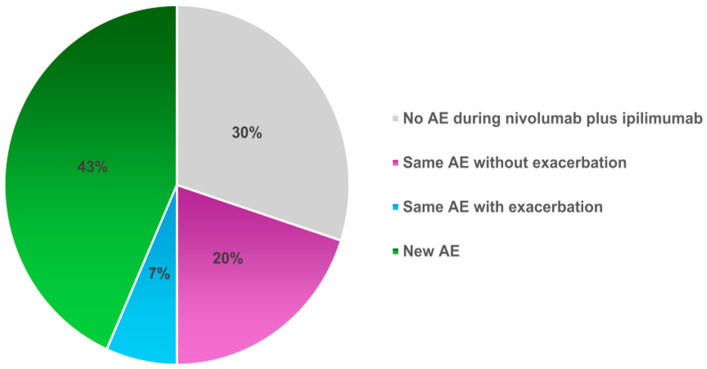
Comparison of AEs during durvalumab with those during nivolumab plus ipilimumab treatment. The percentages of patients treated with nivolumab plus ipilimumab were divided into those who did not experience the same AEs observed during treatment with durvalumab, those who experienced the same AEs, and those who experienced new AE. AE, adverse events.

**Table 1 cancers-16-01409-t001:** Patient characteristics.

		Total *n*, 30 (%) (Range)
Age	Median (range) year	72 (50–80)
Sex	Male/Female	29/1 (96.7/3.3)
History of smoking	Former/Never	29/1 (96.7/3.3)
Histologic type	SQ/AD/NOS	15/14/1 (50/46.7/3.3)
ECOG-PS	0/1/2	10/19/1 (33.3/63.3/3.3)
PD-L1 status	<1/1–49/≥50/Unknown	9/14/5/2 (30/46.7/16.7/6.7)
Pattern of recurrence	Locally/Distant/Locally + Distant	9/13/8 (30/43.3/26.7)
Site of distant metastasis	Brain/Bone/Lung/Liver/Others	9/5/2/2/6 (30/16.7/6.7/6.7/20)
Time from completion of irradiation to recurrence	Median (range) months	6.8 (1.7–30.7)
Dose number of dur	Median (range) times	10.5 (1–26)
Treatment duration of dur	Median (range) months	5.8 (0.03–11.9)
Treatment		
Nivo + ipi without chemotherapy		26 (86.7)
Nivo + ipi + CBDCA + PTX		3 (10)
Nivo + ipi + CBDCA + PEM		1 (3.3)

AD, adenocarcinoma; SQ, squamous cell carcinoma; NOS, not otherwise specified; ECOG-PS, Eastern Cooperative Oncology Group performance status; PD-L1, programmed death-ligand 1; dur, durvalumab; nivo, nivolumab; ipi, ipilimumab; CBDCA, carboplatin; PEM, pemetrexed; and PTX, paclitaxel.

**Table 2 cancers-16-01409-t002:** Tumor response.

	Total *n*, 30	95% CI
PR	7	
SD	14	
PD	9	
ORR	23.3%	12–41
DCR	70%	52–83

PR, partial response; SD, stable disease; PD, progression of disease; ORR, overall response rate; DCR, disease control rate; and CI, confidential interval.

**Table 3 cancers-16-01409-t003:** Univariate and multivariate analyses of predictive factors related to PFS and OS.

PFS								
Factor			Univariate			Multivariate		
		Median PFS	HR	95% CI	*p*-Value	HR	95% CI	*p*-Value
Age	<75/≥75	4.6/3.5	1	0.4–3.0	0.99			
Sex	Male/Female	4.2/4.6	0.7	0.1–13.1	0.77			
ECOG-PS	0/1	NR/5.2	0.7	0.2–2.0	0.53			
Histology	Sq/Non-Sq	6.1/4.1	1.0	0.4–2.6	0.99			
PD-L1	<1/≥1	NR/3.5	0.4	0.1–1.1	0.06	0.6	0.2–1.8	0.38
	<50/≥50	6.1/3.3	0.6	0.2–2.1	0.39			
Brain metastasis	No/Yes	4.6/4.1	1	0.4–2.8	0.97			
Low-dose CBDCA	No/Yes	6.1/3.5	0.8	0.3–2.5	0.67			
Dur start within 14 days of completion of irradiation.	Yes/No	6.4/4.1	1	0.4–2.7	0.99			
Treatment cycles of dur	≥11/<11	NR/3.3	0.3	0.1–0.7	0.005			
Completion of dur	Yes/No	NR/3.5	0.5	0.2–1.2	0.12			
Treatment duration of dur (months)	≥6/<6	NR/3.3	0.3	0.1–0.7	0.005	0.3	0.1–1	0.04
Pattern of recurrence	Locally only/Distance	NR/4.1	0.5	0.1–1.4	0.20	1	0.3–3.4	0.97
Time to recurrence from final CCRT (months)	≥6/<6	NR/3.5	0.4	0.1–0.9	0.04			
OS								
Factor			Univariate			Multivariate		
		Median OS	HR	95% CI	*p*-value	HR	95% CI	*p*-value
Age	<75/≥75	18.5/8.4	0.6	0.2–2.5	0.4	0.6	0.2–2.5	
Sex	Male/Female	18.5/8.0	0.4	0.1–6.9	0.41	0.4	0.1–6.9	
ECOG-PS	0/1	18.5/NR	1.7	0.5–5.2	0.4	1.7	0.5–5.2	0.24
Histology	Sq/Non-Sq	NR/18.5	1.8	0.6–6.1	0.33	1.8	0.6–6.1	
PD-L1	<1/≥1	NR/18.5	0.5	0.1–1.7	0.28	0.5	0.1–1.7	
	<50/≥50	18.5/NR	2.8	0.5–50.4	0.26	2.8	0.5–50.4	0.08
Brain metastasis Low-dose CBDCA	No/Yes	18.5/NR	0.9	0.2–2.7	0.82	0.9	0.2–2.7	
	No/Yes	NR/18.5	0.7	0.2–2.7	0.6	0.7	0.2–2.7	
Dur start within 14 days of completion of irradiation.	Yes/No	18.5/NR	1.5	0.5–5.5	0.5	1.5	0.5–5.5	
Treatment cycles of dur	≥11/<11	NR/7.3	0.2	0.04–0.6	0.005	0.2	0.04–0.6	
Completion of dur	Yes/No	18.5/8.4	0.2	0.03–0.8	0.02	0.2	0.03–0.8	
Treatment duration of dur (months)	≥6/<6	NR/7.3	0.2	0.04–0.6	0.005	0.2	0.04–0.6	0.0007
Pattern of recurrence	Locally only/Distance	18.5/NR	0.6	0.1–1.8	0.35	0.6	0.1–1.8	
Time to recurrence from final CCRT (months)	≥6/<6	18.5/8.2	0.4	0.1–1.3	0.13	0.4	0.1–1.3	

HR, hazard ratio; CI, confidence interval; NR, not reached; ECOG-PS, Eastern Cooperative Oncology Group performance status; Sq, squamous carcinoma; PD-L1, programmed death-ligand 1; CBDCA, carboplatin; dur, durvalumab; and CCRT, concurrent chemoradiotherapy.

**Table 4 cancers-16-01409-t004:** IrAEs of nivolumab plus ipilimumab.

Variable	All Grade *n*, (%)	≥Grade 3 *n*, (%)
Hyper or hypothyroidism	5 (16.7)	0
Adrenal insufficiency	1 (3.3)	0
Pneumonitis	3 (10)	1 (3.3)
Liver dysfunction	6 (20)	0
Diarrhea	2 (6.7)	1 (3.3)
Skin disorder	6 (20)	1 (3.3)
Arthralgia	3 (10)	0
Stomatitis	1 (3.3)	0
Shingles	1 (3.3)	0
Eosinophilia	1 (3.3)	0
γ-GTP increased	1 (3.3)	0
ALP increased	1 (3.3)	0

IrAE, immune-related adverse events; γ-GTP, ϒ-Glutamyl transpeptidase; and ALP, alkaline phosphatase.

## Data Availability

All study data are included in the article.

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
