# Peer review of "Clinical Outcome of Nivolumab Plus Ipilimumab in Patients with Locally Advanced Non-Small-Cell Lung Cancer with Relapse after Concurrent Chemoradiotherapy followed by Durvalumab"

_cancers, 2024, doi:10.3390/cancers16071409_

Round 1
Reviewer 1 Report
Comments and Suggestions for Authors
This is a clearly developed and clinically relevant study on nivolumab/ipilimimab combinations for patients initially suffering from locally-advanced NSCLC who underwent after chemo-radiotherapy and durvalumab. The clinical question is extremely timely.
I have only minor suggestions
In the abstract, it is not clear if the reported outcomes refer to the previous administration of CCCRT-durvalumab or nivo/ipi (this latter being the treatment of interest).
“Immune checkpoint inhibitors (ICIs) are anticancer drugs that may help achieve a long 46
tail plateau.” I agree this sentence is adequate for ICI description, but I do not think it is the right one for starting this study. I suggest to remove it.
I am not familiar with the concept of “treatment load” present in the results, could Authors explain?
“Of 30 patients, 26 patients received nivolumab plus ipilimumab without chemotherapy, while 4 patients received nivolumab plus ipilimumab in combination with platinum-doublet chemotherapy”. Can Authors clarify if the regimens (combinations and doses of the drugs) were the ones described in Checkmate-227 (ipi + nivo) and CheckMate-9LA (ipi + nivo + 2 courses of chemo), or otherwise?
Paragraph 3.2: I would reinforce the concept that these are data corresponding to ipi + nivo, in the very first sentence of the paragraph.
“The median time to treatment failure WAS 5.4 months”
Author Response
This is a clearly developed and clinically relevant study on nivolumab/ipilimumab combinations for patients initially suffering from locally-advanced NSCLC who underwent after chemo-radiotherapy and durvalumab. The clinical question is extremely timely.
I have only minor suggestions
1.In the abstract, it is not clear if the reported outcomes refer to the previous administration of CCCRT-durvalumab or nivo/ipi (this latter being the treatment of interest).
Thank you for your advice. We have revised abstract according to this reviewer’s comment as follows.
“Median PFS and OS with nivolumab plus ipilimumab were 4.2 months (95% confidence interval [CI]: 0.7-7.7) and 18.5 months (95% CI: 3.5-33.5), respectively.”
2.“Immune checkpoint inhibitors (ICIs) are anticancer drugs that may help achieve a long tail plateau.” I agree this sentence is adequate for ICI description, but I do not think it is the right one for starting this study. I suggest to remove it.
We agree with the reviewer’s comment. We have removed the statement “Immune checkpoint inhibitors (ICIs) are anticancer drugs that may help achieve a long tail plateau.”, and revised following sentence to “ICIs are utilized as the standard systemic therapy in various types of cancer” on page 2, lines 46-47.
- I am not familiar with the concept of “treatment load” present in the results, could Authors explain?
We apologize for the confusing wording. We have revised the statement “treatment load of durvalumab” on page 4, line 142 to “number of administrations of durvalumab”.
- “Of 30 patients, 26 patients received nivolumab plus ipilimumab without chemotherapy, while 4 patients received nivolumab plus ipilimumab in combination with platinum-doublet chemotherapy”. Can Authors clarify if the regimens (combinations and doses of the drugs) were the ones described in Checkmate-227 (ipi + nivo) and CheckMate-9LA (ipi + nivo + 2 courses of chemo), or otherwise?
Thank you for your advice. We added CheckMate 227 regimen and CheckMate 9LA regimen following to nivolumab plus ipilimumab and nivolumab plus ipilimumab in combination with platinum-doublet chemotherapy on page 4, lines 146-148.
- Paragraph 3.2: I would reinforce the concept that these are data corresponding to ipi + nivo, in the very first sentence of the paragraph.
“The median time to treatment failure WAS 5.4 months”
Thank you for your comment. We agree with the reviewer’s suggestion. We have revised the statement on page 4, line 153-155 as follows.
“The median PFS and OS with nivolumab plus ipilimumab were 4.2 months (95% CI: 0.7-7.7) and 18.5 months (95% CI: 3.5-33.5) respectively. The median time to treatment failure were 5.4 months (95% CI: 0-11.1) (Figure 1).”
Reviewer 2 Report
Comments and Suggestions for Authors
NICE WORK,CONGRATS!FEW THINGS TO TAKE UNDER CONSIDERATION:1)30 CASES ARE NOT ENOUGH TO SUPPORT THIS KIND OF RESULTS,2)YOU SHOULD EXPLAIN BETTER WHY AND WHEN YOU USED 22C3(sp263 IS LOCKED WITH DURVALUMAB).3)EXPLAIN BETTER THE TYPE OF THE NSCLC(SQ/ADENO).
Author Response
NICE WORK,CONGRATS!FEW THINGS TO TAKE UNDER CONSIDERATION:
- 30 CASES ARE NOT ENOUGH TO SUPPORT THIS KIND OF RESULTS
We agree with the reviewer’s comment. We have stated the relatively small number of patients in this study as limitation on page 9, lines 268-269. We and others are planning to conduct prospective trials to investigate the combination of nivolumab and ipilimumab in NSCLC patients whose cancer progresses during or after durvalumab maintenance therapy.
- YOU SHOULD EXPLAIN BETTER WHY AND WHEN YOU USED 22C3(sp263 IS LOCKED WITH DURVALUMAB)
Thank you for your valuable comment. In previous clinical trials investigating durvalumab, PD-L1 expression in tumor microenvironment was evaluated using SP263 clone. However, due to the high correlation between PD-L1 expression data obtained with 22C3 and SP263, and the popularity of 22C3 in PD-L1 testing, PD-L1 expression has usually been assessed with 22C3 in Japan. We added this statement on page 9, lines 293-298.
- EXPLAIN BETTER THE TYPE OF THE NSCLC(SQ/ADENO).
Table 1 presents the number of patients with lung adenocarcinoma, lung squamous cell carcinoma and not otherwise specified. To make this clear, we added the sentence on page 3, lines 135-137 as follows. “Fifteen patients (50%) had lung squamous cell carcinoma, 14 (46.7%) had adenocarcinoma, and 1 (3.3%) was diagnosed as not otherwise specified.”
Reviewer 3 Report
Comments and Suggestions for Authors
The authors reported the clinical outcome, meaning PFS and OS, for IO-IO combination (nivolumab plus ipilimumab) in LA-NSCLC patients with relapse after durvalumab following concurrent CRT.
Despite the small size of the retrospective study, 30 patients, the results are addressing the question if IO-IO combination could be used after ICI treatment.
Here are my comments:
1. Neither NCCN nor ESMO guidelines recommend this subsequent treatment for relapse in LA-NSCLC patients, after durvalumab consolidation.
Taking into consideration that this kind of therapy approach is prohibited from reimbursement in many Western countries, even if PD-L1 inhibitor and anti-CTLA4 inhibitor in monotherapy might be used subsequently in specific cancer sites (e.g. melanoma), the authors should provide a reassuring statement that this small-size retrospective study is not conclusive for routine clinical practice. The authors should address the reading audience that this study is not an indication of usage and the national and guidelines regulations should apply in each and every case.
2. I am interested if there are LA-NSCLC patients treated with other ICIs (single therapy Nivolumab or Pembrolizumab or Atezolizumab) when relapse after Durvalumab consolidation? If yes, what are the results? If yes, this means that the study has a major bias of selecting and publishing just the positive results?
3. Why the authors selected just almost 18 months? Why early reporting was a priority? "Although the observation period was short, a tail 275 plateau was observed with this treatment modality, and early reporting was a priority."
Author Response
The authors reported the clinical outcome, meaning PFS and OS, for IO-IO combination (nivolumab plus ipilimumab) in LA-NSCLC patients with relapse after durvalumab following concurrent CRT.
Despite the small size of the retrospective study, 30 patients, the results are addressing the question if IO-IO combination could be used after ICI treatment.
Here are my comments:
- Neither NCCN nor ESMO guidelines recommend this subsequent treatment for relapse in LA-NSCLC patients, after durvalumab consolidation. Taking into consideration that this kind of therapy approach is prohibited from reimbursement in many Western countries, even if PD-L1 inhibitor and anti-CTLA4 inhibitor in monotherapy might be used subsequently in specific cancer sites (e.g. melanoma), the authors should provide a reassuring statement that this small-size retrospective study is not conclusive for routine clinical practice. The authors should address the reading audience that this study is not an indication of usage and the national and guidelines regulations should apply in each and every case.
Thank you for your valuable advice. We added following sentences according to this comment on page 9, lines 276-280.
“This study suggested the potential efficacy of nivolumab plus ipilimumab combination therapy in patients with recurrent NSCLC following durvalumab consolidation therapy. However, it should be noted that adherence to clinical guidelines and regulatory agency policies is imperative when considering its clinical use.”
- I am interested if there are LA-NSCLC patients treated with other ICIs (single therapy Nivolumab or Pembrolizumab or Atezolizumab) when relapse after Durvalumab consolidation? If yes, what are the results? If yes, this means that the study has a major bias of selecting and publishing just the positive results?
Thank you for your comment. In Japan, the administration of ICI monotherapy to recurrent NSCLC patients after durvalumab consolidation therapy is rare, and currently, there is no available data on patients treated with ICI monotherapy following durvalumab.
- Why the authors selected just almost 18 months? Why early reporting was a priority? "Although the observation period was short, a tail plateau was observed with this treatment modality, and early reporting was a priority."
Thank you for your valuable comment. In the Pacific study, despite of the use of durvalumab following CCRT, 45% of cases experienced recurrence within one year. Therefore, we believe it is advisable to generate data showing the efficacy of nivolumab plus ipilimumab in these patients as soon as possible. To make this clear, we added sentences on page 9, lines 281-285 as follows.
“Although the Pacific study demonstrated a significant improvement in survival with durvalumab consolidation therapy, approximately 45% of patients experienced recurrence after CCRT-durvalumab within one year. Therefore, we believe it is advisable to generate data demonstrating the efficacy of nivolumab plus ipilimumab in these patients as soon as possible.”
Round 2
Reviewer 3 Report
Comments and Suggestions for Authors
The authors revised and corrected the mentioned comments from the first revision.
The manuscript in this form is accepted for publication.